# Bioprospecting, Synergistic Antifungal and Toxicological Aspects of the Hydroxychalcones and Their Association with Azole Derivates against *Candida* spp. for Treating Vulvovaginal Candidiasis

**DOI:** 10.3390/pharmaceutics16070843

**Published:** 2024-06-21

**Authors:** Lígia de Souza Fernandes, Letícia Sayuri Ogasawara, Kaila Petronila Medina-Alarcón, Kelvin Sousa dos Santos, Samanta de Matos Silva, Letícia Ribeiro de Assis, Luís Octavio Regasini, Anselmo Gomes de Oliveira, Maria José Soares Mendes Giannini, Maria Virginia Scarpa, Ana Marisa Fusco Almeida

**Affiliations:** 1Laboratory of Clinical Mycology, Department of Clinical Analysis, School of Pharmaceutical Sciences, São Paulo State University (UNESP), Road Araraquara Jaú, Km 01, S/N, Araraquara 14800-903, SP, Brazil; ligia.fernandes@unesp.br (L.d.S.F.); ls.ogasawara@unesp.br (L.S.O.); kaila.medina@unesp.br (K.P.M.-A.); k.santos@unesp.br (K.S.d.S.); samanta.matos@unesp.br (S.d.M.S.); maria.giannini@unesp.br (M.J.S.M.G.); 2Department of Drugs and Medicines, School of Pharmaceutical Sciences, São Paulo State University (UNESP), Road Araraquara Jaú, Km 01, S/N, Araraquara 14800-903, SP, Brazil; anselmo.gomes@unesp.br (A.G.d.O.); virginia.scarpa@unesp.br (M.V.S.); 3Department of Chemistry and Environmental Sciences, Institute of Biosciences, Humanities and Exact Sciences, São Paulo State University (UNESP), St. Quirino de Andrade, 215, São José do Rio Preto 01049-010, SP, Brazil; leticia.assis@unesp.br (L.R.d.A.); luis.regasini@unesp.br (L.O.R.)

**Keywords:** vulvovaginal candidiasis, *Candida* spp., hydroxychalcone, *Galleria mellonella*, lipid carriers, toxicity

## Abstract

Vulvovaginal candidiasis (VVC) remains a prevalent fungal disease, characterized by challenges, such as increased fungal resistance, side effects of current treatments, and the rising prevalence of non-*albicans Candida* spp. naturally more resistant. This study aimed to propose a novel therapeutic approach by investigating the antifungal properties and toxicity of 2-hydroxychalcone (2-HC) and 3′-hydroxychalcone (3′-HC), both alone and in combination with fluconazole (FCZ) and clotrimazole (CTZ). A lipid carrier (LC) was also developed to deliver these molecules. The study evaluated in vitro anti-*Candida* activity against five *Candida* species and assessed cytotoxicity in the C33-A cell line. The safety and therapeutic efficacy of in vivo were tested using an alternative animal model, *Galleria mellonella*. The results showed antifungal activity of 2-HC and 3′-HC, ranging from 7.8 to 31.2 as fungistatic and 15.6 to 125.0 mg/L as fungicide effect, with cell viability above 80% from a concentration of 9.3 mg/L (2-HC). Synergistic and partially synergistic interactions of these chalcones with FCZ and CTZ demonstrated significant improvement in antifungal activity, with MIC values ranging from 0.06 to 62.5 mg/L. Some combinations reduced cytotoxicity, achieving 100% cell viability in many interactions. Additionally, two LCs with suitable properties for intravaginal application were developed. These formulations demonstrated promising therapeutic efficacy and low toxicity in *Galleria mellonella* assays. These results suggest the potential of this approach in developing new therapies for VVC.

## 1. Introduction

Vulvovaginal candidiasis (VVC) is the opportunistic infection of the female reproductive tract mucous membrane caused by exacerbated growth of yeast belonging to the *Candida* genus, which causes changes in vaginal discharge, irritation, itching, and a burning sensation [1].

According to epidemiological data, about 75% of women present at least one episode of VVC during their lives, and 9% develop recurrent vulvovaginal candidiasis that consists of three or more episodes of VVC in one year [2,3,4]. It is considered a significant public health problem in all regions of the world, responsible for high morbidity rates and compromising the performance of women’s normal daily activities in addition to physical and emotional discomfort and a negative financial burden [5,6].

*Candida albicans* is the principal etiological agent of VVC. About 90% of the cases are caused by this species, followed by *C. tropicalis*, *C. glabrata*, and *C. parapsilosis* [7,8,9,10,11,12]. The publication made by the World Health Organization in 2022 shows that *Candida* infections still deserve much attention, and further studies are needed concerning some *Candida* species [13].

Treatment of vaginal candidiasis is carried out mainly with antifungals of the azole and polyene classes, but cases of fungal resistance have been frequently observed [14]. One of the VVC challenges is related to the increase in *C. albicans* resistance (acquired resistance) to some drugs and the emergence of non-*albicans* species, which are intrinsically more resistant to some drugs beyond their mechanisms of acquired resistance [10,12,15,16,17].

Therefore, antifungal therapies, after oral administration, are also limited due to their toxicity, side effects in pregnant women, and drug interactions. For this reason, topical treatments are often more advantageous than oral drugs [18,19,20]. Local administration of antimicrobials has also been considered to allow greater control of drug administration and reduce or avoid undesirable systemic adverse effects [20,21].

The exposed facts justified the search for new molecules and potential therapeutic strategies for treating infections caused by *Candida*. New therapeutic agents, such as essential oils, probiotics, antimicrobial peptides, and natural active principles of plant origin, are rich sources of new molecules with diverse pharmacological activities and reduced toxic effects [22] and can be combined with conventional medicines.

Hydroxychalcones are the main subclass of the chalcone group (a class of polyphenolic compounds that are precursors in the biosynthesis of flavonoids) obtained from the hydroxylation of the aromatic rings of the chalcone molecule [23]. Hydroxychalcones 2-hydroxychalcone (2-HC) and 3′-hydroxychalcone (3′-HC) can be found naturally or cheaply synthesized quickly in the laboratory. Some of the pharmacological activities of these two molecules have already been reported, and our research group has already demonstrated their antifungal activity [24,25,26]. Recent studies in pharmaceutical technology have focused on the development of drug delivery systems capable of improving the distribution and retention of drugs in the vaginal mucous membrane, providing greater patient acceptability, and a sustained release of the drug also, such as increased activity and reduced toxicity [20,27].

In addition, they can also be used as potentiators of some drugs through synergistic interactions between them, which can lead to an increase in the spectrum of action, as well as in therapeutic efficiency, reduce drug toxicity, and prevent the emergence of resistant strains. Many synergistic effects between natural compounds and conventional drugs have been demonstrated and have shown high efficacy in treating multifactorial diseases (cancer, diabetes, and others) and infectious diseases [28,29,30,31,32]. However, no data is available on the susceptibility profiles of *Candida* spp. to combined 2-HC and 3′-HC with azole derivatives, along with the drug delivery system, capable of improving the distribution and retention of drugs in the vaginal mucous membrane for a sustained release of the drug; the same happens with increased activity and reduced toxicity.

Thus, the present study aimed to explore two hydroxychalcones, their associations with azole drug activity against *Candida* species, and their toxicological safety. It also aimed to develop a lipid carrier capable of solubilizing and transporting these molecules with low aqueous solubility.

## 2. Materials and Methods

### 2.1. Chemicals

The 2-hydroxychalcone (2-HC) and 3′-hydroxychalcone
(3′-HC) (Figure 1) were synthesized using Claisen–Schmidt condensation with 2′-hydroxyacetophenone and benzaldehyde in the basic medium, as outlined by Zeraik et al. (2012) in collaboration with Prof. Dr. Luis Octávio Regasini’s research group from the Institute of Biosciences, Letters and Exact Sciences-UNESP, Brazil [33]. These compounds were identified by 1H-nuclear magnetic resonance and 13C (1H- and 13C-NMR), in which 1H-NMR displayed a coupling constant range (J) of H-α and H-β of 15.0 Hz, corresponding to an (E)-diastereomer [33,34]. Fluconazole (FCZ), clotrimazole (CTZ), and amphotericin B (AMB) were obtained from Sigma-Aldrich. 

### 2.2. Yeasts

*Candida albicans ATCC 90028, Candida tropicalis* ATCC 750, *Candida krusei* ATCC 6258, *Candida glabrata* ATCC 90030, and *Candida parapsilosis* ATCC 22019 were used in this study. The yeasts were obtained from the Clinical Mycology Laboratory of the Department of Clinical Analysis, Faculty of Pharmaceutical Sciences-UNESP, Brazil. Microorganisms were grown on Sabouraud dextrose agar (Acumedia, Lansing, MI, USA) and incubated at 37 °C for 24 h [35].

### 2.3. In Vitro Susceptibility of Candida spp.

Susceptibility tests were performed according to the M27-A3 protocol proposed by the Clinical & Laboratory Standards Institute (CLSI) (2008) [35]. Briefly, 2-HC and 3′-HC were solubilized in dimethyl sulfoxide at 12.500 g/L and stored at −80 °C. The working solutions were prepared in Roswell Park Memorial Institute (RPMI)-1640 medium with L-glutamine, without sodium bicarbonate, and with phenol red as the pH indicator (Gibco^®^, Grand Island, NY, USA) and buffered with 4-Morpholinepropanesulfonic acid hemisodium salt (MOPS) (Sigma-Aldrich, Saint Louis, MO, USA), pH = 7.2. The final concentration of 2-HC and 3′-HC in the RPMI medium was 0.24–125.0 mg/L. Dilutions of the compounds were performed in a 96-well microplate at a total volume of 100 μL/well. Fungal suspensions were prepared in 0.85% NaCl and were adjusted to a final concentration of 5 × 10^3^ cells/mL. Then, 100 μL of the fungal suspension was added to all wells. Microplates were incubated at 37 °C for 24 h. Visual and colorimetric readings were performed by adding 20 μL of 0.01% resazurin. FCZ (0.125–64.0 mg/L; to *C. krusei*: 0.25–128.0 mg/L) and AMB (0.016–8.0 mg/L) were used as methodology controls and in addition, the minimum inhibitory concentration of CTZ was also tested in the range of 0.008–4.0 mg/L for *C. albicans*, *C. krusei* and *C. tropicalis* and 0.004–2.0 mg/L for *C. glabrata* and *C. parapsilosis*. Minimal inhibitory concentration (MIC) was considered the lowest concentration of the drug that inhibited the visible growth of yeast [35,36,37]. To determine the minimum fungicide concentration (MFC), a sample from each well that showed antifungal activity was added to Petri dishes containing Sabouraud dextrose agar (BD Difco™, Franklin Lakes, NJ, USA) and incubated at 37 °C for 24 h. Samples were processed in triplicate, and each experiment was carried out in triplicate.

### 2.4. In Vitro Cytotoxicity

Cytotoxicity assays were performed on the C33A cell line (uterine cervical carcinoma epithelium) obtained from the ATCC (Manassas, VA, USA). 5 × 10^4^ cells/well were added using RPMI medium with 10% fetal bovine serum into 96-well plates [38]. Cells were incubated at 36.5 °C with 5% CO_2_ for 24 h to form a cell monolayer. 2-HC and 3′-HC were analyzed in range concentrations from 0.1–75 μg/mL. In combinations, 2-HC and 3′-HC were analyzed from 0.48–31.25 mg/L and 0.48–62.5 mg/L. Cells were exposed for 72 h, and after, cytotoxicity was assessed by adding 100 μL of resazurin (50 μM) and measured by a microplate reader at 570 nm–600 nm. A negative control with untreated cells (100% living cells) and a positive control (DMSO 40% to induce cell death) were also used [26,39]. The results were evaluated in triplicate independent. The inhibition concentration 50% (IC_50_) value was defined as the highest drug concentration at which 50% of the cells are viable relative to the control, and it was obtained by a dose–response curve using GraphPad Prism 5.0. Selective index (SI) was calculated using SI=IC50/MIC50. Despite these promising results, the selectivity index (SI) values were below the SI ≥ 10 threshold, considered the minimum value for drug safety [40].

### 2.5. Combinatorial Antifungal Activity

*Candida albicans* ATCC 90028, *Candida tropicalis* ATCC 750, and *Candida krusei* ATCC 6258 were used. These species were selected for these assays because *C. albicans* is the most prevalent etiological agent in fungal infections caused by the genus *Candida,* and *C. tropicalis* is the most frequent non-*albicans* species. *C. krusei* was used to represent the species most resistant to azole drugs. Thus, 2-HC and 3′-HC were combined with FCZ and CTZ and associated with each other. Their combinatorial antifungal activity was determined through the checkerboard method using 96-well microplates, according to Bellio et al. (2021) [41]. Concentration ranges evaluated were as follows: 2-HC (0.06–62.50 mg/L) + FCZ (0.063–4.0 mg/L), 3′-HC (0.06–62.50 mg/L) + FCZ (0.063–4.0 mg/L), 2-HC (0.06–62.50 mg/L) + CTZ (0.002–0.125 mg/L), 3′-HC (0.06 a 62.50 mg/L) + CTZ (0.002–0.125 mg/L), and 3′-HC (0.49–31.25) + 2-HC (0.06 a 62.50 mg/L). Microplates were incubated at 37 °C for 24 h. Then, 20 μL of resazurin (0.01% in water) was added to each well. The plates were incubated at 37 °C ± 1 °C for 3 h, and the percentages of fungal growth inhibition were determined using a microplate reader at 570 nm. The negative control was only the culture medium, and the positive control was composed of microorganisms and the culture medium. The percentages of inhibition of fungal growths were calculated as follows:$$\mathrm{Inhibition}\;\left(\%\right)=1-\left(\frac{\mathrm{Absorbance}-\mathrm{Negative}\;\mathrm{control}\;\mathrm{absorbance}}{\mathrm{Positive}\;\mathrm{control}\;\mathrm{absorbance}-\mathrm{Negative}\;\mathrm{control}\;\mathrm{aborbance}}\right)\times 100$$

For the interpretation of the checkerboard, the “Lowest Fractional Inhibitory Concentration Index” (FICI) method of each well of the growth and non-growth interface, as proposed by Feng et al. (2021) and Feng and Yang (2023) [42,43]. The wells with more than 80% fungal growth inhibition were used to limit the interface. From this, it was calculated the fractional inhibitory concentration (FIC) and the fractional inhibitory concentration index (FICI) of each well of the interface as follows:$$\mathrm{FIC}\;=\left(\mathrm{MIC}80\;\mathrm{compound}\;\mathrm{in}\;\mathrm{combination}/\mathrm{MIC}80\;\mathrm{compound}\;\mathrm{alone}\right)$$
FICI = FIC compound 1+FIC compound 2

The minor FICI obtained was used according to classification: synergistic (FICI ≤ 0.5), additive (0.5 < FICI ≤ 1.0), indifferent (1.0 < FICI < 4.0), and antagonistic (FICI ≥ 4.0) [44].

### 2.6. In Vitro Cytotoxicity of the Combinations

The cytotoxicity of the combinations was also evaluated; after the monolayer formation, the active ingredients were combined in the microplate according to Bellio et al. (2021) in the same concentrations described above [41]. The combinations’ cell viability percentages were evaluated punctually, i.e., the percentages of cell viability corresponding to the concentrations of the wells with the lowest FICI obtained in the combinatorial antifungal activity assays were expressed.

### 2.7. Development of Lipid Carrier (LC)

Studies were conducted to develop a system capable of acting as a vehicle for the drugs studied in this work. For this, a ternary phase diagram was constructed composed of capric/caprylic triglyceride (Captex 300, Abitec Corporation, Columbus, OH, USA—oil phase), soybean phosphatidylcholine (Epikuron 200, Hamburg, Germany), and hydrogenated castor oil (Croduret 50^®^, Croda, Princeton, NJ, USA) (surfactant and co-surfactant in the ratio 25:75 m/m) and water [45]. Each mixture corresponding to each point of the diagram was kept in an ice bath and under ultrasonic stirring (QSonica Sonicators with standard probe, Model Q700—maximum power output of 700 watts, Newtown, CT, USA) in a discontinuous mode for 5 min (1 min of sonication and 20 s intervals) and an amplitude of 1. After 24 h, the systems were visually classified as LVTS: low viscosity and translucent system; LVOS: low viscosity and opaque system; MVTS: medium viscosity and translucent system; MVOS: medium viscosity and opaque system; HVTS: high viscosity and translucent system; HVOS: high viscosity and opaque system; and PS: phase separation. From this classification, different regions were delimited in the pseudo-ternary phase diagram.

#### 2.7.1. Mechanical Characterization of LC

##### Texture Profile Analysis

The texture profile of the LC was evaluated using the TA-XT plus texture analyzer (Stable Micro Systems, Surrey, UK). For that, 7.0 g of formulation were weighed in a conical tube. The sample was centrifuged at 3500 rpm for 5 min and left to rest for 24 h. Subsequently, the samples were kept below the analytical probe at 37 °C, and the following parameters were used: 10 mm analytical probe, 5 mm distance, compression and return speed of 0.5 mm/s, and a second compression after 5 s. This assay was also performed with LC containing 2-HC (0.1%) and with a commercial formulation (clotrimazole vaginal cream 1%—Medley, Campinas, Brazil), which contained benzyl alcohol, cetostearyl alcohol, sorbitan stearate, cetyl palmitate, polysorbate 60, triglycerides of capric and caprylic acids, and purified water.

##### In Vitro Determination of Mucoadhesive Strength

Texture analyzer TA-XT plus (Stable Micro Systems, Surrey, UK) was employed in adhesion test mode. We used mucin discs of 0.8 mm in diameter, prepared from a solution of mucin at 5% (*w*/*w*) in phosphate-buffered saline (PBS) and simulated vaginal fluid (FVS) maintained at 37 °C, as proposed by Owen and Katz (1999) [46]. A total of 7.0 g of LC was weighed in a conical tube in a bath at 37 °C to evaluate the mucoadhesive strength. The samples are centrifuged at 3500 rpm for 5 min and left to rest for 24 h. These discs were fixed to the lower end of the analytical probe of the equipment (10 mm probe), and then 20 μL of simulated vaginal fluid was added. The analytical probe was moved with a speed of 1 mm/s until the mucin disc was close to the sample’s surface. After contact of the discs with the samples for 20 s, the probe rose (speed of 0.5 mm/s) until there was detachment between the disc and the sample. The force required to highlight the samples in contact with the mucin discs (mucoadhesive force) was calculated by the force versus time curve. In addition, the mucoadhesive strength of the LC containing 0.1% of 2-HC and the commercial formulation (clotrimazole vaginal cream 1%—Medley, Campinas, Brazil) was also analyzed.

#### 2.7.2. Analytical Method

In this study, 2-HC was quantified in the formulations by UV-vis spectrophotometry. A Shimadzu Mini-1280 UV–visible (Columbia, MD, USA) spectrophotometer was used, and the cuvette used was quartz, with an optical path of 1 cm. The standard and sample solutions were diluted in ethanol and analyzed at 352 nm. An analytical curve was constructed at 6.03–14.06 mg/L concentrations.

#### 2.7.3. Encapsulation Efficiency

The capacity of the developed system to solubilize 2-HC was evaluated; for this, an excess of 2-HC (2.0%) was added to 1.0 g of formulation, and the system was homogenized. Then, the formulation was centrifuged at 14,000 rpm for 3 h. After this period, 200 mg of the formulation, located at the top of the tube, was weighed and solubilized in ethanol at the theoretical final concentration of 10.04 mg/L. The amount of 2-HC incorporated into the formulation was obtained according to the analytical method described, and the percentage of recovery of the analyte in the formulation as follows [47]:$$\mathrm{Recovery}\;\left(\%\right)=\frac{\mathrm{Experimental}\;\mathrm{concentration}\;\left(\mathrm{mg}/L\right)}{\mathrm{Theoretical}\;\mathrm{concentration}\;\left(\mathrm{mg}/L\right)}$$

### 2.8. Toxicity and Efficacy in Galleria mellonella Model

Larvae (225–275 mg) without dark spots were selected and maintained in Petri dishes (*n* = 8 for the group) in the dark at 37 °C and without food the night before experiments. In toxicological evaluation tests, the larvae were inoculated with 10 µL of 2-HC, 3′-HC, and their associations in PBS and LC solutions in PBS (0.25 g of the LC in 200 µL of PBS). Then, the larvae were incubated at 37 °C, and survival was assessed daily for five days [48]. A group of larvae that received PBS injection was used as a control. In vivo, assays of antifungal activity (therapeutic efficacy) were conducted by infecting larvae with 10 µL of *C. albicans* ATCC 90028 inoculum (5 × 10^5^ CFU/larva) in left proleg after being cleaned with 70% ethanol [49]. After 1 h of infection, the larvae were injected into the last right proleg with 10 µL of 2-HC, 3′-HC, associations, and LC solutions in PBS (0.25 g of the LC in 200 µL of PBS). The control groups used were as follows: (1) A group of infected larvae and treated with FCZ (350 mg/L or 17.5 mg/kg) [50] was used as a treatment control group; (2) Uninfected larvae and treated with PBS; (3) Infected larvae without treatment as an infection control group. Larvae were incubated at 37 °C and evaluated for five-day survival [48]. The experiment was repeated twice, and in both trials, the larvae of *G. mellonella* were also monitored by assessing the following parameters: mobility, cocoon formation, melanization, and survival, as proposed by Loh et al. (2013) with modifications. A healthy, uninfected wax worm score is between 9 and 10, and an infected, dead wax worm scores 0 [51].

## 3. Results

### 3.1. In Vitro Susceptibility of Candida spp

The results of antifungal activities of 2-HC, 3’-HC, CTZ, FCZ, and AMB against *Candida* spp are shown in Table 1. The 2-hydroxychalcone (2-HC) showed MIC of 31.3, 31.3, 7.8, 15.6, and 7.8 mg/L for *C. albicans, C. tropicalis, C. krusei, C. glabrata,* and *C. parapsilosis*, respectively. The 3′-hydroxychalcone (3′-HC) showed MICs of 31.3, 31.3, 7.8, 7.8, and 15.6 mg/L for *C. albicans, C. tropicalis, C. krusei, C. glabrata*, and *C. parapsilosis*, respectively. The best MFC of 2-HC was for *C. parapsilosis* (15.6 mg/L) and *C. krusei* (125.0 mg/L), while for 3′-HC, *C. krusei* (31.3 mg/L), *C. glabrata* (31.3 mg/L), *C. parapsilosis* (31.3 mg/L), and *C. tropicalis* (125.0 mg/L) (Table 1).

### 3.2. In Vitro Cytotoxicity

Following the ISO EN 10993-5 protocol criteria [52], our results demonstrated a cytotoxic effect at the highest concentrations tested of both substances. However, this effect was attenuated at the concentration of 9.38 mg/L, reaching cell viability above 80% for both substances, 2-HC and 3′-HC, showing a significant difference of (*p* < 0.0001) when compared with the highest concentrations (Figure 2). The MIC_50_ values ranged from 15.8 to 7.8 for *Candida* spp., and the IC_50_ values were 12.97 mg/L for 2-HC and 11.86 mg/L for 3′-HC. The selectivity indexes (SI) ranged from 0.83 to 1.66 for 2-HC (0.83 to *C. albicans* and *C. krusei*, 1.66 to *C. tropicalis*) and 0.76 to 1.52 for 3′-HC (0.76 to *C. albicans*, 1.52 to *C. tropicalis* and *C. krusei*). These results can be seen in Table 2.

### 3.3. In Vitro Combinatorial Antifungal Activity

The antifungal activity of 2-HC and 3′-HC was also evaluated in combination with azole drugs (FCZ and CTZ) and combined with each other. The results obtained in the assessment of the combinatorial antifungal activity and the lowest FICI are shown in Table 3, and as can be observed, the combination of 3′-HC + 2-HC and 2-HC + FCZ presented lower FICI means with values of 1.8 and 1.4, respectively. The combination between the two chalcones was synergistic for all *Candida* species, while the other combinations varied depending on the drug’s association and species. The minor FICI obtained was used to classify the interactions, as shown in Table 4.

### 3.4. In Vitro Cytotoxicity of the Combinations

Combinations of 2-HC and 3′-HC with azole derivatives (FCZ and CTZ) were evaluated for cytotoxicity in their MIC_80_ values against strains of *C. albicans*, *C. tropicalis,* and *C. krusei* (Table 5). The results showed that the combinations generally presented high cell viability. The 3′-HC + FCZ combination showed maximum cell viability of 100% in the MICs for *C. albicans* and *C. tropicalis*. They were followed by the combination of 2-HC + FCZ, which achieved cell viability of 94.92% and 98.35% in the MICs of the evaluations against *C*. *albicans* and *C. tropicalis,* respectively. The combination 2-HC + CTZ showed a cell viability of 95.04% and 100% in the MIC of *C. albicans* and *C. krusei*, respectively. The combination 3′-HC + CTZ showed 90.60% cell viability to *C. krusei* MIC. Finally, the combinations 3′-HC + 2-HC showed 86.91% and 75.63% viability in the MICs of *C. albicans* and *C. krusei*, respectively. The SI of 2-HC and 3′-HC were also calculated in the combinations with FCZ and CTZ (Table 6), and high levels of SI were obtained, depending on the MIC of each species. The highest value achieved was the combination 2-HC + CTZ with an SI of 382.16 in the MICs of *C. albicans* and *C. krusei*, as well as 3′-HC + FCZ that presented a value equal to 241.67 for *C. albicans* and *C. tropicalis*; thus, some combinations showed high levels of selectivity.

### 3.5. Development of Lipid Carrier (LC)

Figure 3 shows the visual characteristics and classifications of the formulations obtained, and Figure 4 depicts the pseudo-ternary phase diagram obtained from these classifications. Two systems were selected for evaluating the solubilization and carrier capacity of the drugs: LC (1) comprised 30% surfactant/co-surfactant (75:25, *w*/*w*), 35% oil phase, and 35% water (high viscosity and translucent system—purple area in Figure 4), and LC (2) consisted of 30% surfactant/co-surfactant (75:25, *w*/*w*), 22% oil phase, and 48% water (high viscosity and translucent system—purple area in Figure 4).

### 3.6. Mechanical Characterization of LCs and Encapsulation Efficiency

The mechanical properties of LC (1) and LC (2), empty and containing 1.0% of 2-HC, were evaluated, and the results are shown in Figure 5. Comparisons were made with the mechanical properties of a commercial formulation of clotrimazole 1%. Higher and statistically significant values were observed for the properties of hardness, compressibility, and cohesion of the systems developed concerning the commercial formulation (*p* < 0.05). For adhesion, LC (1), LC (1) + 2-HC (0.1%), LC (2), LC (2) + 2-HC (0.1%), and clotrimazole 1% (commercial) presented means and standard deviations in values equal to 5.68 ± 1.02, 7.9 ± 2.17, 4.94 ± 0.48, 4.64 ± 1.01, and 0.56 ± 0.07 respectively. Thus, the two lipid carriers (empty and containing 2-HC) were more adhesive than the commercial formulation, but LC (1) showed a more significant statistical difference with the commercial formulation compared to LC (2). In the in vitro mucoadhesive strength evaluation assay, empty LC (2) containing 2-HC showed values equal to 0.71 ± 0.15 and 0.84 ± 0.14, respectively. These values were higher aesthetically significant strength compared to the clotrimazole 1% (commercial) (0.21 ± 0.05), LC (1) (0.21 ± 0.05), and LC (1) + 2-HC (0.1%) (0.19 ± 0.11).

Regarding the solubilization and incorporation efficiency of 2-HC, LC (1) achieves a recovery percentage of 92.96% ± 1.75, while LC (2) demonstrates 104.28% ± 0.58. Then, both carriers exhibit high incorporation efficiency.

### 3.7. Evaluation of Toxicological and Therapeutic Efficacy In Vivo Model Galleria mellonella

The doses administered in vivo were determined from the MIC_80_ values obtained in vitro combinatorial antifungal assays against *C. albicans* (Table 7) and according to the solubility of the active ingredients in PBS. A group of larvae that received only PBS injection was used as a control group. For the combination of the 3′-HC + 2-HC has been observed 100% live larvae and health scores above eight for all doses tested, indicating healthy larvae and absence of toxicity and no statistically significant difference about the control group in the values of health indexes (*p* < 0.05) (Figure 6A,B). The 2-HC + FCZ also showed 100% survival except at 97.6 mg/L concentration. However, there was also no statistically significant difference with the control group (*p* = 0.2046, log-rank [(Mantel–Cox]) test) (Figure 6C). As for the health indexes, the group that received the doses 0.244 + 0.125 e 97.6 + 50.0 presented statistical difference values concerning the control group (*p* < 0.05) (Figure 6D). Finally, for groups that have received the substances alone (at their respective MIC_80_ values) and for LC (1) and LC (2) has been observed 100% live larvae (Figure 6E), and high health scores in all groups showing the absence of toxicity (Figure 6F) and no statistical difference to the control group (*p* < 0.05). The absence of toxicity of the lipid carriers was considered relevant since the formulations do not present toxicity, which suggests biocompatibility and safety for pharmaceutical use [53,54].

In the trials evaluating the therapeutic efficacy in *G. mellonella*, a control group of larvae received only the injection of the inoculum of *C. albicans* ATCC 90028 (5 × 10^5^ CFU/larva). All survival curves showed statistical differences concerning the control, demonstrating the doses’ effectiveness (Figure 7A,C). However, regarding health indexes, the combination of 2-HC + 3′-HC did not show a statistical difference with the control group (untreated larvae) in any of the doses (Figure 7B). The combination of 2-HC + FCZ also showed a statistical difference compared to the control group, demonstrating the effectiveness of the doses in this model, as shown in Figure 7C. Regarding the health index for 2-HC + FCZ, only the dose of 48.8 + 25.0 slightly showed a statistical difference compared to the control group of infected larvae without any treatment. As expected, the treatment control group (FCZ 350 mg/L) was statistically different from the infection control group (Figure 7B,D) in survival curves and health indexes. Figure 7E,F demonstrate that 2-HC and 3′-HC alone and the formulations did not demonstrate efficacy in the antifungal activity against the evaluated *C. albicans.*

## 4. Discussion

Bioprospecting research often uses substances from plant biodiversity to search for value-added bioproducts with potential clinical applications [55,56,57,58]. Chalcones (1,3-diaryl-2-propen-1-ones), for example, are part of a class of bioactive compounds belonging to the flavonoid family and have great pharmacological potential, such as antioxidant, antineoplastic, and antimicrobial activity, and for the treatment of cardiovascular diseases [59,60,61,62,63,64]. In previous results, 2-chalcone-mediated photodynamic therapy was found to be a promising and safe drug candidate against dermatophytes, particularly in anti-biofilm treatment [25], and 3′-hydroxychalcone. This molecule also demonstrated antimicrobial and mixed antibiofilm activity against *Mycobacterium tuberculosis* and *Paracoccidioides brasiliensis*, with MIC values of 7.8 and 0.97 mg/L and presented biofilm degradation above 50%, showing itself to be a promising substance in the evaluation of these microorganisms [26].

In this context, we explored the antifungal potential of hydroxychalcones (2-HC and 3′-HC). The high incidence of infections caused by *Candida* spp, the increase in fungal resistance to available drugs, and the increased prevalence of non-*albicans* species, which are intrinsically more resistant to drugs, were focal points for this work [65]. Both showed antifungal activity against *Candida* spp. strains and fungicidal activity against *C. krusei, C. glabrata,* and *C. parapsilosis* strains. The results obtained demonstrate a highly interesting potential against these strains. Thus, the results presented here corroborate previous studies that indicate chalcones as a promising class in developing prototypes aimed at antifungal therapy. Fungicidal activity is essential because substances that kill pathogens are strong candidates for clinical use [66]. Kant et al. (2016) revealed the potential antifungal activity of 25 chalcones against *Candida* spp., and the MICs obtained were 6.25–50.0 mg/L. *C. albicans* was particularly susceptible to compounds in the chalcone class, with presenting MICs ranging from 1.0 to 64.0 mg/L [67].

In the cytotoxicity assessment assays, the C33A cell line was chosen because it represents, although it is a tumoral line, the target anatomical site of the study, which is the vaginal epithelium. Our results also showed that 2-HC and 3′-HC showed cytotoxicity at high concentrations (above 9.38 mg/L). The SI showed values below 10; due to this result, we evaluated the need to combine both molecules with each other and the existing drugs (FCZ and CTZ) as new therapeutic strategies. Furthermore, *Candida* infections have a high incidence, and FCZ is usually used to treat these infections. However, its widespread and indiscriminate use has led to a significant problem with fungal resistance and therapeutic failures [68]. Combined therapies have gained a lot of interest and act by rescuing the activity of drugs through association with other molecules to circumvent these problems [68,69,70,71].

Recent studies described the flavonoid’s antifungal activity with FCZ, and studies by Wang et al. (2016) and Chai et al. (2023) showed several chalcones with synergistic antifungal activity with FCZ. Chai et al. (2023) described the synthesis of quinolone-chalcone derivatives, and their results demonstrated that most derivatives exhibited good antifungal activity against *C. albicans* resistant to FCZ, restoring susceptibility to this drug [70,71,72].

Our work also addressed evaluating the antifungal activity of combinations of hydroxychalcones with azole derivatives. The following species were selected for the combinatorial antifungal activity assays: *C. albicans*, *C. tropicalis,* and *C. krusei. C. albicans* is responsible for the most infections, and *C. tropicalis* and *C. krusei* were selected as the non-*albicans* species intrinsically resistant to azole drugs. The MIC_80_ was established to provide greater precision of antifungal activity. All the combinations demonstrated promising antifungal results, highlighting the combinations of 2-HC + 3′-HC, with MIC values of 3.9 to 7.8 mg/L improving antifungal activity when compared to the individual effect and, in turn, demonstrated a synergistic effect against all three of the *Candida* species observed. On the other hand, the 2-HC + FCZ combination demonstrated high antifungal activity, with MICs of 15.6 to 0.06 mg/L, in addition to exhibiting a synergistic *C. krusei* and additive effect against *C. tropicalis*, species genetically resistant to FCZ [43,65].

The evaluation of cytotoxicity of the combinations was evaluated for 2-HC + FCZ, 2-HC + CTZ, 3′-HC + FCZ, and 3′-HC + CTZ because, in these cases, both FCZ and CTZ did not present cytotoxicity in the concentration ranges tested, so the cytotoxic effect evaluated was only due to hydroxychalcones in these combinations. Thus, the IC_80_ was calculated based on the concentrations of hydroxychalcones since the dose–response curve represents the effect of a single drug as a function of its concentration. Considering the cytotoxicity of hydroxychalcones, it was impossible to establish this correlation for the combination of 2-HC + 3′-HC.

Significant reductions in MICs led to an increase in cell viability and SI values above 10. The combinations could reduce cytotoxicity and maintain antifungal activity [40,73].

Another alternative was to develop a lipid carrier for intravaginal application to reduce the cytotoxicity and to deliver the drugs. In the development of the formulation, a pseudo-ternary phase diagram was constructed. Visual classifications of the samples prepared by mixing different proportions of aqueous, oily, and surfactant/co-surfactant phases enabled the delineating distinct regions in a pseudo-ternary phase diagram. These systems were chosen based on high viscosity to ensure prolonged drug retention at the site of action and high concentrations of the oil phase and surfactant/co-surfactant, considering the limited aqueous solubility of hydroxychalcones [74,75].

Mechanical properties of the topical formulations directly influence clinical efficacy in sensory parameters, acceptability of the product, and treatment adherence [76,77,78,79]. These parameters are also correlated with stability and internal organization within systems. Hardness and compressibility indicate the force required to induce deformation, which is associated with the formulation’s ease of application and spreadability on the biological surface. Adhesion and cohesion represent the energy needed to break attractive forces between the sample surface and the analytical probe [76,77,78,79].

Compared to the commercial formulation, the notably high hardness and compressibility obtained signify rigid internal connections within the systems, suggesting enhanced stability and internal organization, making them more resistant to deformation, properties that are important to intravaginal application. LC (1) and LC (2) also exhibit elevated cohesion and adhesion compared to the commercial formulation, indicating possible structural recovery post-application [80].

Evaluation of in vitro mucoadhesive strength demonstrated that empty LC (2) and LC (2) + 2-HC (0.1%) showed in vitro mucoadhesive strength that was statistically significant and superior concerning the commercial formulation, which demonstrates greater interaction with the fluids of the infected vaginal mucous membrane, facilitating sustained presence at the application site, which could enhance clinical efficacy in future applications. LC (1), empty and containing 0.1% 2-HC, showed higher adhesion compared to LC (2), LC (2) containing 0.1% of 2-HC and commercial formulation; this adhesion reflects more the attractive forces within the system itself and not the interaction with the vaginal mucous membrane as in the mucoadhesive strength assessment assay [81]. Finally, regarding the efficiency of incorporation, the formulation proved to be adequate for transporting 2-HC, showing potential for intravaginal application.

Continuing our research, we evaluated the toxicity of the combinations 3′-HC + 2-HC and 2-HC + FCZ, which gave the best synergistic results in our research, as well as the lipid carriers LC (1) and LC (2) in *G. mellonella*, according to the survival curve in the five-day evaluation of the experiment. Survival results of 100% were shown in all concentrations for both combinations 3′-HC + 2-HC and 2-HC + FCZ. The parameters related to the health of the larvae, such as lack of movement, melanization, decreased cocoon formation, and death, can be used as markers of diseases in *G. mellonella* larvae evaluated through the health index [51]. Maximum values of six and eight for the combination of 3′-HC + 2-HC and 2-HC + FCZ, as well as isolated substances and for the lipid carriers LC (1) and LC (2), were observed, demonstrating that these combinations did not show toxicity and that they may be promising in the treatment of VVC. Similar studies have demonstrated the effectiveness of the agreement model in the evaluation of drugs combined with fluconazole (FCZ) against *C. albicans* [82].

*G. mellonella* is a valuable model for evaluating nanostructured formulations and systems’ toxicity and therapeutic efficacy, aligning with experimental animals’ reduction, refinement, and replacement principles [54,80,83]. Therefore, it is an ideal platform for toxicity screening assays, biocompatibility, and efficacy assessment of antifungal drugs. Although the proposed route of administration for the developed LCs is local, their injectable application in *G. mellonella* larvae allows the assessment of systemic toxicity [54,80,84].

Therefore, we consider the results necessary because *G. mellonella* larvae is a more complex model concerning in vitro assays, and none of the combinations and doses tested showed toxicity. The combinations and doses tested in the therapeutic efficacy trials also showed antifungal activity against *C. albicans*. It was impossible to perform the in vitro assays using the lipid carriers with the incorporated chalcones due to the composition and consistency of the lipid carriers, so we chose to perform them in an alternative animal model of *G. mellonella*. However, we are still developing more appropriate models for evaluating formulations containing hydroxychalcones and trials to evaluate therapeutic efficacy by topical administration would perhaps be more appropriate.

## 5. Conclusions

Our study contributes to the discovery of new combined drugs for use in treating CVV. Our results demonstrated that the combinations of 2-HC and 3′-HC with azole derivatives (FCZ and CTZ) demonstrated a solid antifungal action against *C. albicans* and non-*albicans* strains, with high cell viability in vitro assays mainly in the combinations and without toxicity, and with effective action in the in vivo treatment of *G. mellonella*. Combinations of 3′-HC + 2-HC and 2-HC + FCZ can be used as promising therapeutic approaches. In addition, our results demonstrated the possibility of obtaining lipid carriers with potential application in clinical practice and contributing to developing an effective formulation for treating CVV. However, further studies will be needed to understand the mechanism of action of these combinations and better understand the molecular bases of these combinations.

## Figures and Tables

**Figure 1 pharmaceutics-16-00843-f001:**
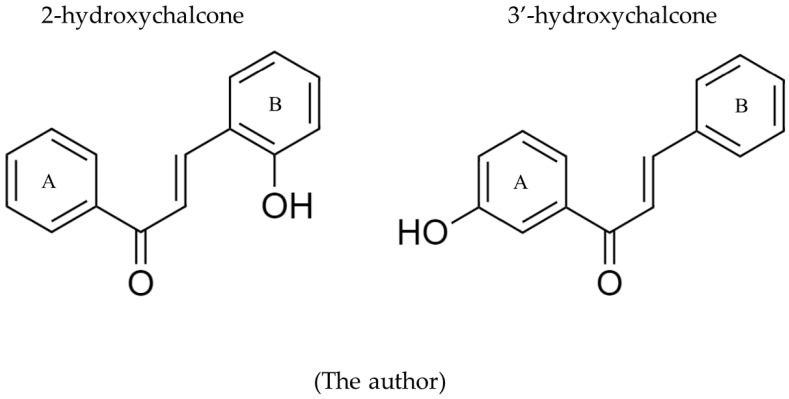
Structural formulas of 2-hydroxychalcone (2-HC) and 3′-hydroxychalcone (3′-HC).

**Figure 2 pharmaceutics-16-00843-f002:**
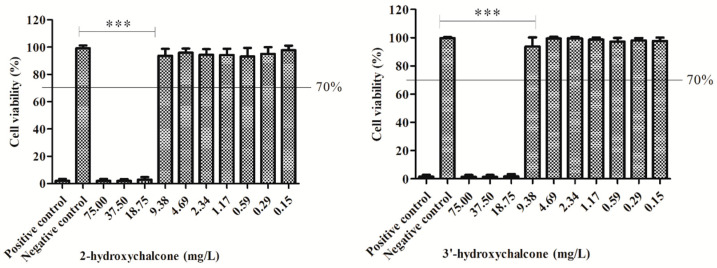
Percentage of cell viability after exposure to different concentrations of 2-hydroxychalcone and 3′-hydroxychalcone in cell line: C33A. Positive control: DMSO 40% (cell death). Negative control: 100% living cells. The results are expressed as mean ± standard deviation one-way ANOVA followed by Bonferroni (*p* < 0.001). *** Statistically significant difference, the assays were performed from three replicates of three independent experiments.

**Figure 3 pharmaceutics-16-00843-f003:**
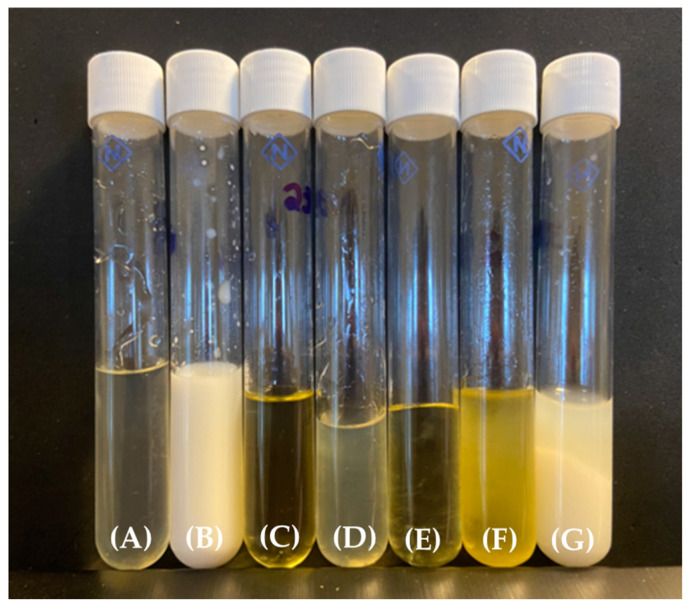
Visual classification of the systems: (**A**)—low viscosity and translucent system; (**B**)—low viscosity and opaque system; (**C**)—medium viscosity and translucent system; (**D**)—medium viscosity and opaque system; (**E**)—high viscosity and translucent system; (**F**)—high viscosity and opaque system; (**G**)—phase separation.

**Figure 4 pharmaceutics-16-00843-f004:**
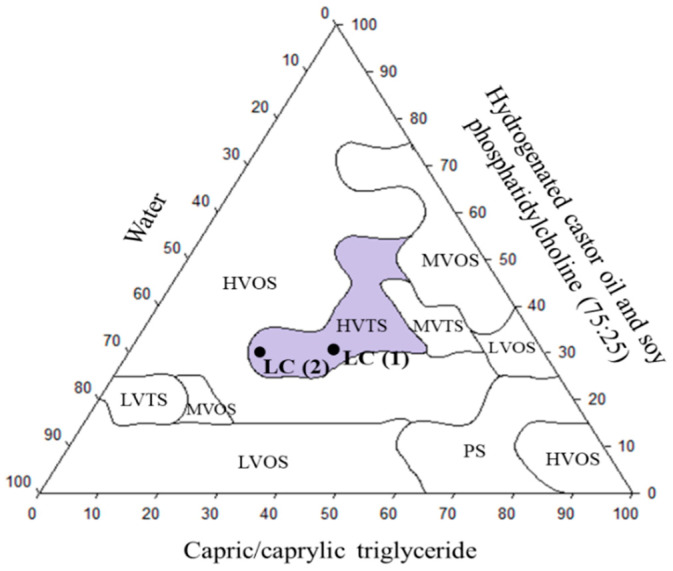
Pseudo-ternary phase diagram of hydrogenated castor oil and soy phosphatidylcholine (75:25) (surfactant/co-surfactant), capric/caprylic triglyceride (oily phase), and water. LC (1) and LC (2) were the selected formulations for mechanical characterization and encapsulation efficiency. LVTS: low viscosity and translucent system, LVOS: low viscosity and opaque system, MVTS: medium viscosity and translucent system, MVOS: medium viscosity and opaque system, HVTS: high viscosity and translucent system, HVOS: high viscosity and opaque system, PS: phase separation.

**Figure 5 pharmaceutics-16-00843-f005:**
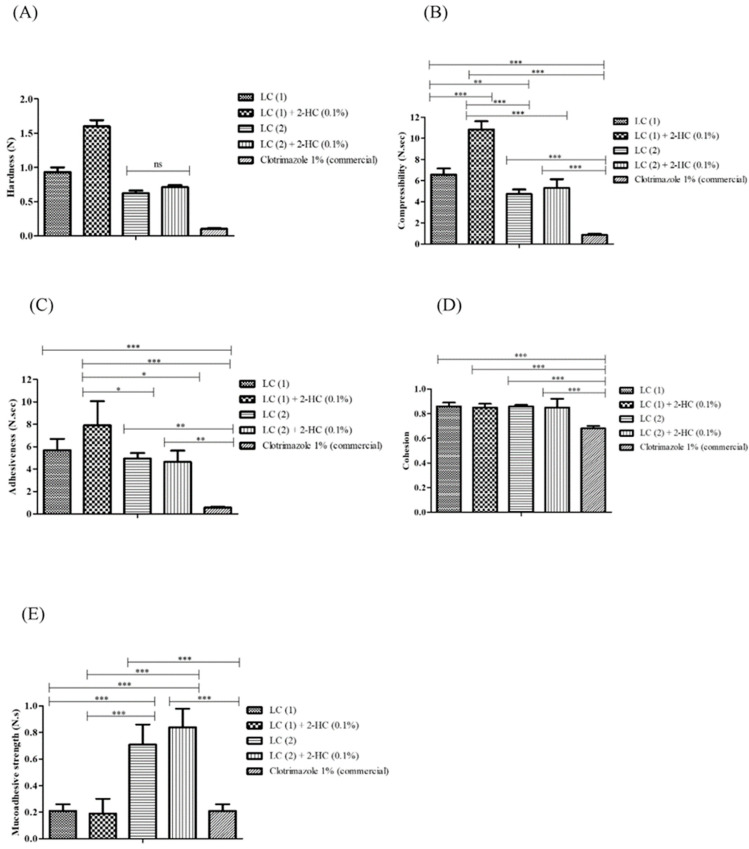
Evaluation of the mechanical properties of empty lipid carriers (LC1 and LC2), lipid carriers containing 1.0% of 2-HC, and clotrimazole 1% (commercial vaginal cream). (**A**): hardness (N); (**B**): compressibility (N.sec); (**C**): adhesiveness (N.sec); (**D**): cohesion; (**E**): mucoadhesive strength (N.s). Mean ± SD. One-way ANOVA followed by Bonferroni’s post hoc comparison tests were used in all statistical analyses (*p* < 0.05). ns; Not significant. LC (1): Lipid carrier composed of 30% surfactant/co-surfactant (75:25, *w*/*w*), 35% oil phase, and 35% water. LC (2): lipid carrier composed of 30% surfactant/co-surfactant (75:25, *w*/*w*), 22% oil phase, and 48% water. 2-HC: 2-hydroxychalcone. Tests performed on independent quadruplicates. *** *p* < 0.001. ** 0.001 < *p* < 0.01. * 0.01 < *p* < 0.05.

**Figure 6 pharmaceutics-16-00843-f006:**
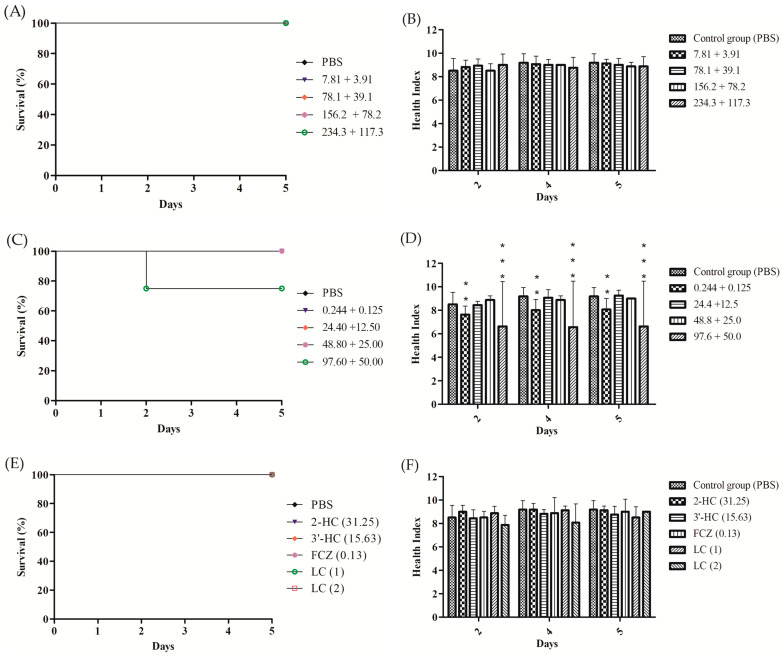
Kaplan–Meier survival curves and health index after application of different concentrations (mg/L) of the combination of 3′-HC + 2′-HC (**A**,**B**), 2-HC + FCZ (**C**,**D**), and the MIC values obtained and LCs solutions (**E**,**F**) in the *G. mellonella.* Statistical analyses to survival curves: log-rank (Mantel–Cox) Test (*p* < 0.001). * Survival curve statistically different compared to the control group (infection). Health index: results expressed as mean ± standard deviation. One-way ANOVA followed by Bonferroni. *** *p* < 0.001. ** 0.001 < *p* < 0.01. Experiments were performed twice.

**Figure 7 pharmaceutics-16-00843-f007:**
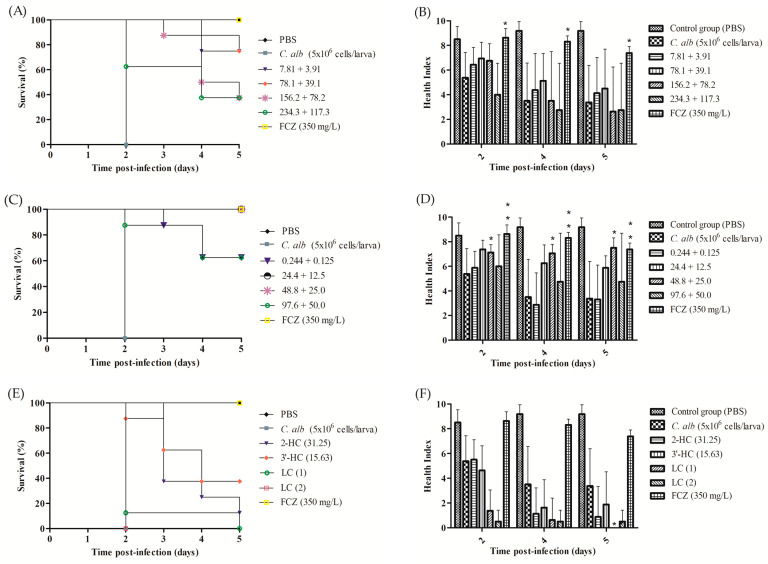
Kaplan–Meier survival curves and health index after infection (5 × 10^6^ cells/larva—*C. albicans* ATCC 90028) and treatment with different concentrations (mg/L) of the combination of 3′-HC + 2-HC (**A**,**B**), 2-HC + FCZ (**C**,**D**) and of the substances at the respective MIC values obtained and of the LCs (**E**,**F**). Statistical analyses to survival curves: log-rank (Mantel–Cox) test (*p* < 0.001). Health index: results are expressed as mean ± standard deviation. One-way ANOVA followed by Bonferroni. ** 0.001 < *p* < 0.01. * 0.01 < *p* < 0.05. Experiments were performed twice. (**F**): * Health index = 0.

**Table 1 pharmaceutics-16-00843-t001:** Summary of fungistatic and fungicidal effects of 2-hydroxychalcone (2-HC), 3′-hydroxychalcone (3′-HC), clotrimazole (CTZ), fluconazole (FCZ), and amphotericin B (AMB) against reference strains (ATCC) of different *Candida* species.

	Yeasts									
	*C. albicans* (ATCC 90028)		*C. tropicalis* (ATCC 750)		*C. krusei* (ATCC 6258)		*C. glabrata* (ATCC 90030)		*C. parapsilosis* (ATCC 22019)	
	MIC	MFC	MIC	MFC	MIC	MFC	MIC	MFC	MIC	MFC
2-HC	31.3	>125.0	31.3	>125.0	7.8	125.0	15.6	>125.0	7.8	15.6
3′-HC	31.3	>125.0	31.3	125.0	7.8	31.3	7.8	31.3	15.6	31.3
CTZ	0.01	>4.0	0.01	>4.0	0.06	>4.0	0.004	>2.0	0.03	0.1
FCZ	0.25	>64.0	0.50	>64.0	32.0	128.0	0.50	>64.0	1.00	2.0
AMB	0.20	1.0	0.05	1.0	0.10	1.0	0.40	0.5	0.10	0.3

MIC: minimum inhibitory concentration (mg/L); MFC: minimum fungicidal concentration (mg/L); All tests were performed in three independent experiments.

**Table 2 pharmaceutics-16-00843-t002:** Minimum inhibitory concentration 50% (MIC_50_), inhibition concentration 50% (IC_50_), and selectivity (SI) index of 2-hydroxychalcone (2-HC) and 3′-hydroxychalcone (3’-HC).

	2-HC			3′-HC		
	MIC_50_ (mg/L)	IC_50_ (mg/L)	SI	MIC_50_ (mg/L)	IC_50_ (mg/L)	SI
*C. albicans* ATCC 90028	15.63	12.97	0.83	15.63	11.86	0.76
*C. tropicalis* ATCC 750	7.81		1.66	7.81		1.52
*C. krusei* ATCC 6258	15.63		0.83	7.81		1.52

**Table 3 pharmaceutics-16-00843-t003:** In vitro interaction between 2-hydroxychalcone (2-HC) and 3′-hydroxychalcone (3’-HC) with fluconazole (FCZ), clotrimazole (CTZ), and combined each other.

		**MIC_80_ 2-HC** **(individual)**	**MIC_80_ 2-HC** **(combination)**	**FIC 2-HC**	**MIC_80_ FCZ** **(individual)**	**MIC_80_ FCZ** **(combination)**	**FIC FCZ**	**FICI**
2-HC + FCZ	*C. albicans*	31.25	0.24	0.01	0.50	0.13	0.25	0.3
	*C. tropicalis*	31.25	0.06	0.002	0.50	0.50	1.00	1.0
	*C. krusei*	31.25	15.63	0.50	32.00	0.50	0.02	0.5
		**MIC_80_ 2-HC** **(individual)**	**MIC_80_ 2-HC** **(combination)**	**FIC 2-HC**	**MIC_80_ CTZ** **(individual)**	**MIC_80_ CTZ** **(combination)**	**FIC CTZ**	**FICI**
2-HC + CTZ	*C. albicans*	31.25	0.06	0.002	0.03	0.02	0.50	0.5
	*C. tropicalis*	31.25	15.63	0.50	0.03	0.002	0.06	0.6
	*C. krusei*	31.25	0.06	0.002	0.13	0.13	1.00	1.0
		**MIC_80_ 3′-HC** **(individual)**	**MIC_80_ 3′-HC** **(combination)**	**FIC 3′-HC**	**MIC_80_ FCZ** **(individual)**	**MIC_80_ FCZ** **(combination)**	**FIC FCZ**	**FICI**
3′-HC + FCZ	*C. albicans*	31.25	0.06	0.002	0.50	0.25	0.50	0.5
	*C. tropicalis*	62.50	0.06	0.001	0.50	0.25	0.50	0.5
	*C. krusei*	15.63	0.06	0.00	32.00	32.00	1.00	1.0
		**MIC_80_ 3′-HC** **(individual)**	**MIC_80_ 3′-HC** **(combination)**	**FIC 3′-HC**	**MIC_80_ CTZ** **(individual)**	**MIC_80_ CTZ** **(combination)**	**FIC CTZ**	**FICI**
3′-HC + CTZ	*C. albicans*	31.25	15.63	0.50	0.03	0.002	0.06	0.6
	*C. tropicalis*	62.50	62.50	1.00	0.03	0.002	0.06	1.1
	*C. krusei*	15.63	7.81	0.50	0.13	0.01	0.06	0.6
		**MIC_80_ 3′-HC** **(individual)**	**MIC_80_ 3′-HC** **(combination)**	**FIC 3′-HC**	**MIC_80_ 2-HC** **(individual)**	**MIC_80_ 2-HC** **(combination)**	**FIC 2-HC**	**FICI**
3′-HC + 2-HC	*C. albicans*	31.25	7.81	0.25	31.25	3.91	0.13	0.4
	*C. tropicalis*	31.25	7.81	0.25	62.50	15.63	0.25	0.5
	*C. krusei*	15.63	3.91	0.25	31.25	7.81	0.25	0.5

FIC: fractional inhibitory concentration. FICI: fractional inhibitory concentration index. MIC_80_: minimum inhibitory concentration of 80% (mg/L) of the individual drugs and their combination.

**Table 4 pharmaceutics-16-00843-t004:** Lower fractional inhibitory concentration index and interactions obtained from combinations of 2-hydroxychalcone (2-HC) and 3′-hydroxychalcone (3’-HC) with azole derivatives (fluconazole—FCZ and clotrimazole—CTZ) and associated with each other.

Associations	Yeast		
	*C. albicans*	*C. tropicalis*	*C. krusei*
2-HC + FCZ	0.3	1.0	0.5
	Synergistic	Additive	Synergistic
2-HC + CTZ	0.5	0.6	1.00
	Synergistic	Synergistic (partial)	Additive
3′-HC + FCZ	0.5	0.5	1.00
	Synergistic	Synergistic	Additive
3′-HC + CTZ	0.6	1.1	0.6
	Synergistic (partial)	Indifferent	Synergistic (partial)
3′-HC + 2-HC	0.4	0.5	0.5
	Synergistic	Synergistic	Synergistic

**Table 5 pharmaceutics-16-00843-t005:** Concentrations of the combination drugs (mg/L) that presented the lowest FICI in the combinatorial antifungal activity assays, the interactions, and the percentages of cell viability.

2-HC + FCZ		**MIC_80_ 2-HC (combination)**	**MIC_80_ FCZ (combination)**	**Interaction**	**% Cell viability**
	*C. albicans*	0.24	0.13	SYN	94.92
	*C. tropicalis*	0.06	0.50	AD	98.35
	*C. krusei*	15.63	0.50	SYN	38.85
2-HC + CTZ		**MIC_80_ 2-HC (combination)**	**MIC_80_ CTZ (combination)**	**Interaction**	**% Cell viability**
	*C. albicans*	0.06	0.02	SYN	95.04
	*C. tropicalis*	15.63	0.002	SYN (P)	15.19
	*C. krusei*	0.06	0.13	AD	100.00
3′-HC + FCZ		**MIC_80_ 3′-HC (combination)**	**MIC_80_ FCZ (combination)**	**Interaction**	**% Cell viability**
	*C. albicans*	0.06	0.25	SYN	100.00
	*C. tropicalis*	0.06	0.25	SYN	100.00
	*C. krusei*	0.06	32.00	AD	**
3′-HC + CTZ		**MIC_80_ 3′-HC (combination)**	**MIC_80_ CTZ (combination)**	**Interaction**	**% Cell viability**
	*C. albicans*	15.63	0.002	SYN (P)	10.07
	*C. tropicalis*	62.50	0.002	IND	0.00
	*C. krusei*	7.81	0.01	SYN (P)	90.60
3′-HC + 2-HC		**MIC_80_ 2-HC (combination)**	**MIC_80_ 3′-HC (combination)**	**Interaction**	**% Cell viability**
	*C. albicans*	3.91	7.81	SYN	86.91
	*C. tropicalis*	15.63	7.81	SYN	0.00
	*C. krusei*	7.81	3.91	SYN	75.63

** No verified due to the concentration range tested in the combination cytotoxicity assay. 2-HC: 2-hydroxychalcone; 3′-HC: 3′-hydroxychalcone. CTZ: clotrimazole, FCZ: fluconazole. MIC_80_: minimum inhibitory concentration 80%; SYN: synergism; SYN (P): partial synergism; AD: additive; IND: indifferent.

**Table 6 pharmaceutics-16-00843-t006:** Minimum inhibitory concentration (MIC_80_%), inhibition concentration (IC_80_%), and selectivity index (SI) of 2-hydroxychalcone (2-HC) and 3′-hydroxychalcone (3’-HC) and their associations.

		**MIC_80_ 2-HC (combination)**	**IC_80_**	**SI**
2-HC + FCZ	*C. albicans*	0.24	16.96	70.67
	*C. tropicalis*	0.06		282.67
	*C. krusei*	15.63		1.08
		**MIC_80_ 2-HC (combination)**		
2-HC + CTZ	*C. albicans*	0.06	22.93	382.16
	*C. tropicalis*	15.63		1.47
	*C. krusei*	0.06		382.16
		**MIC_80_ 3′-HC (combination)**		
3′-HC + FCZ	*C. albicans*	0.06	14.50	241.67
	*C. tropicalis*	0.06		241.67
	*C. krusei*	0.06		241.67
		**MIC_80_ 3′-HC (combination)**		
3′-HC + CTZ	*C. albicans*	15.63	14.72	0.94
	*C. tropicalis*	62.50		0.23
	*C. krusei*	7.81		1.88

**Table 7 pharmaceutics-16-00843-t007:** Doses administered (concentrations) to larvae (mg/L).

	3′-HC + 2-HC (mg/L)		2-HC + FCZ (mg/L)
3′-HC + 2-HC (MIC_80_ combination)	7.81 + 3.91	2-HC + FCZ (MIC_80_ combination)	0.244 + 0.125
x 10	78.10 + 39.10	x 100	24.40 + 12.50
x 20	156.20 + 78.20	x 200	48.80 + 25.00
x 30	234.30 + 117.30	x 400	97.60 + 50.00

2-HC: 2-hydroxychalcone; 3′-HC: 3′-hydroxychalcone; FCZ: fluconazole. MIC_80_: minimum inhibitory concentration 80%.

## Data Availability

The data presented in this study are available upon request from the corresponding author. The data are not publicly available due to privacy and ethical concerns.

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
