# Peer review of "Bioprospecting, Synergistic Antifungal and Toxicological Aspects of the Hydroxychalcones and Their Association with Azole Derivates against Candida spp. for Treating Vulvovaginal Candidiasis"

_pharmaceutics, 2024, doi:10.3390/pharmaceutics16070843_

Round 1
Reviewer 1 Report
Comments and Suggestions for Authors
General Impression
The authors present the results of a multidisciplinary study on the antifungal effects and toxicity of two related plant secondary metabolites, 2-hydroxychalcone and 3-hydroxychalcone. This study is well designed and includes experiments on synergies with commonly prescribes azole antifungals, drug delivery and systemic effects on waxworm larvae. The introduction summarizes the theoretical framework accurately, using recent and appropriate citations. The experiments are described well and data analysis is appropriate. The paper suffers from a weakness in data interpretation, as the authors do not discuss the suitability of the chalcones for antifungal therapy in detail. This reviewer is concerned that the comparably high toxicity and low antifungal potency, the examined compounds might not be suitable for clinical use. If this were the authors’ interpretation, it would need more emphasis.
Major problem
The major problem with the potential clinical value of the compounds is hinted at in the right column of table 4. In the field of herbal medicine, a selective toxicity index (SI) below 1 suggests that the compound is toxic and can cannot be used as a drug [1]. The chalcones used in this study clearly fall into this category. The manuscript needs to be clearer in pointing this out as this potentially disqualifying property of the compounds is not discussed appropriately.
More details and minor issues
Methods, line 173: Please specify the type of sonicator used – knowing the dial setting on the power output is of limited value if one does not know the instrument.
Methods, line 201: speed of movement is missing the time unit (mm per what?)
Methods, line 214-221: Please rewrite the description of the procedure – I was not able to follow. I was trying to understand how the method can show a recovery above 100% (line 375).
Table 3: I believe the authors want to classify the FICI scores as additive, not as addictive.
Table 5: Still has some Portuguese in it.
1. Indrayanto, G.; Putra, G.S.; Suhud, F. Chapter Six - Validation of in-vitro bioassay methods: Application in herbal drug research. In Profiles of Drug Substances, Excipients and Related Methodology, Al-Majed, A.A., Ed. Academic Press: 2021; Vol. 46, pp. 273-307.
The authors address the issues of whether 2 chalcone components have useful antimicrobial action against Candida species - either alone or in combination with common antifungal agents of the azole class.. This question is addressed in concert with the potential toxicity of these compounds in cell culture as well as in the Galleria model.
The study addresses the gap in knowledge about the potential antifungal action and toxicity of the 2- and 3-OH chalcones.
The study goes in deep detail on the potential synergies between chalcones and azole; in addition, the studies on pharmacological formulations to preserve or enhance the efficacy of the drugs are going beyond previously published studies.
There are no further suggestions about controls or improvements to the methodology. The authors have employed a very broad spectrum of methods and analyses. The data show that chalcones are not very promising for the treatment of fungal infections and further broadening the study to involved testing on mammals is not appropriate.
The authors should address the issue with the low selectivity of chalcone action because this would preclude its use in antifungal therapy. It is my understanding that SIs should be above 10 to warrant further investigation.
The references are current and accurately present the state of knowledge in the field.
Tables and figure are appropriate.
A few minor issues as noted in the comments
Author Response
Answer Revisor 1:
The major problem with the potential clinical value of the compounds is hinted at in the right column of table 4. In the field of herbal medicine, a selective toxicity index (SI) below 1 suggests that the compound is toxic and can cannot be used as a drug [1]. The chalcones used in this study clearly fall into this category. The manuscript needs to be clearer in pointing this out as this potentially disqualifying property of the compounds is not discussed appropriately.
- Answer: Although the chalcones exhibited SI values below the threshold of SI ≥10, which is an established standard for selectivity and safety, we observed potential in these compounds. Therefore, to enhance antifungal activity and cell viability, the chalcones studied were combined with the standard drugs FCZ and CTZ, as well as with each other. The results showed a significant improvement in antifungal activity and a notable reduction in cytotoxicity, achieving 80% cell viability for both 2-hydroxychalcone and 3'-hydroxychalcone. Consequently, this led to improved SI values above SI ≥10, indicating greater selectivity for the microorganism. Previous studies from our group have also reported that combinatorial activities enhance antifungal efficacy and reduce cytotoxicity. For example, we can cite studies that demonstrate these benefits, Soares et al., 2014, which showed the benefits of combining protocatechuates with standard drugs, obtaining pronounced antifungal activity with high selectivity indexes, DOI: 10.1155/2014/957860. Therefore, other studies demonstrate this impact in a similar way DOI: 10.1080/1040841X.2021.1884641, DOI: 10.1080/21505594.2020.1868814, DOI: 10.1007/s10482-009-9394-8
2.Methods, line 173: Please specify the type of sonicator used – knowing the dial setting on the power output is of limited value if one does not know the instrument.
Answer: It has already been added to the line 249
3.Methods, line 201: speed of movement is missing the time unit (mm per what?)
Answer: The fix has been inserted. The mm refers to the speed at which the analytical probe moves and is programmed in millimeters in the equipment to the line 280 and 282.
- Methods, line 214-221: Please rewrite the description of the procedure – I was not able to follow. I was trying to understand how the method can show a recovery above 100% (line 375).
Answer: It was corrected on the Line 296 - 303 and the methodology was rewritten
Table 3: I believe the authors want to classify the FICI scores as additive, not as addictive.
Answer: It was corrected on the Line 386
Table 5: Still has some Portuguese in it.
Answer: We made the requested correction Line 409

Reviewer 2 Report
Comments and Suggestions for Authors
The topic discussed by the authors is very interesting and important due to the large number of infections e.g. vulvovaginal candidiasis caused by Candida and the increasing drug resistance of these fungi. Therefore, the search for a new, low-toxic drugs and an effective ways of delivering these drugs using carriers that can increase the effectiveness of drugs and, for example, reduce their toxicity to the host is very important.
However, I have a few comments for the authors;
1) To determine the cytotoxicity of preparations, the authors should use primarily normal cells, not cancer cells. Cancer cell lines can be used additionally, not alone. Please explain what was the positive control in the study. Moreover, this information should be included in the description of Figure 2. Please complete the caption of the figure so that it is clear what the graphs show without reading the methods, now the caption is unclear. You should add the number of times the experiment was repeated and information about the statistics. Why is 4.69 mg/L given as a non-cytotoxic concentration for 3'-HC, if the graph shows that survival is over 90% at a concentration of 9.38 mg/L, and according to the definition, 70% survival is enough.
2) Please state what statistical test was used in each experiment and add this information in the figure and table captions.
3) The discussion should be strengthened because in the chapter combining the results and discussion, the obtained results and their interpretation are mainly described. However, there is no discussion in the context of other studies.
4) Please check the unit of concentration of the tested substances throughout the work, because sometimes it is given in mg/mL, sometimes in mg/L - it should be standardized.
5) Definitions of the values ​​used should be provided, e.g. SI, FICI, MFC and so on.
6) In materials and methods, point 2.1, the authors write that they used three drugs: FCZ, CTZ, AMB, while in point 2.2.1, lines 129-130, only FCZ and AMB were used to assess the susceptibility of Candida spp. Why was a different FCZ concentration range used for C. krusei than for other Candida? However, in Table 1, the MIC and MFC results of all preparations and drugs are given for all strains. Please check and correct the information in point 2.2.1.
7) Please explain why the antifungal activity of the combined preparations and drugs was assessed only against three out of five Candida species - C. albicans, C. tropicalis, C. krusei. On what basis were they selected? Moreover, why was AMB not used in these studies?
8) Table 4 is the EC50 of 2-HC against the three strains the same as in the case of 3'-OH? Whether different, please indicate this clearly.
9) line 284 what does CIM80 mean? Shouldn't it be MIC80?
10) explanations of abbreviations should be placed below the table and not in the title
11) line 267 - standardize the letter size - uppercase or lowercase, check it out throughout
12) line 221 please complete the method description
13) lines 256-263 the authors describe the results presented in Table 1, therefore line 257 does not present the results from Table 1 but the concentration range used to determine MIC and MFC - please correct. Line 257 this description is incorrect because the table only gives established MIC and MFC values ​​for individual Candida species, and not the range of concentrations used, please change it
14) why the infected insects were given such a high dose of 350 mg/L FCZ (line 231) and Fig 6. On what basis was this concentration chosen, because previously Fig 5 was administered 0.13 mg/L
please explain the difference between MIC (Table 1) and MIC80 (Table 2) as their values ​​are different in the tables. If MIC values were determined for all species, they should be shown for all strains tested. On what basis were only three species C. albicans, C. tropicalis and C. krusei selected for combinatorial testing of antifungal activity? This needs to be justified and why wasn't AMB used in this test if it worked at much lower concentrations than FCZ and CTZ? on what basis were the doses selected for in vivo cytotoxicity testing, shown in fig5E, F
15) line 259 - MIC 3'-HC for C. albicans is 3.25, and according to Table 1 is 31.25 mg/ml - please correct and insert the correct value
16) line 267 explanation of the abbreviations MIC and MFC standardize the letter size - uppercase or lowercase, this applies to the entire text please check and correct it
17) I have reservations to the control in the experiment, because if the groups tested after C. albicans infection were given in a combination of 3’-HC+2-HC with different concentrations or FCZ, i.e. the insects were subjected to double immunization. Why were control larvae immunized only once if the study groups were immunized twice? To correctly interpret the results, you must prepare the appropriate controls.
18) Why, for in vivo toxicity testing on Galleria mellonella larvae, much lower concentrations of 3'-HC and 2-HC were used alone than those administered in combination. On what basis were they selected for experiments? How much LC (1) and LC (2) were administered? How was the dose estimated? On what basis was the dose of C. albicans selected for insect infections? - please complete the information. Why is the health index not shown on day 5 but on day 4, since percent survival was assessed for 5 days. How many times were experiments on insects repeated? There is no statistical evaluation of the obtained results.
Author Response
Answer Revisor 2:
- To determine the cytotoxicity of preparations, the authors should use primarily normal cells, not cancer cells. Cancer cell lines can be used additionally, not alone. Please explain what was the positive control in the study. Moreover, this information should be included in the description of Figure 2. Please complete the caption of the figure so that it is clear what the graphs show without reading the methods, now the caption is unclear. You should add the number of times the experiment was repeated and information about the statistics. Why is 4.69 mg/L given as a non-cytotoxic concentration for 3'-HC, if the graph shows that survival is over 90% at a concentration of 9.38 mg/L, and according to the definition, 70% survival is enough
Answer: Cytotoxicity assays were conducted using the C33-A human cervical carcinoma cell line. Although this cell type is primarily studied in the context of cervical cancer, it was chosen for our study due to its specific application in evaluating vulvovaginal candidiasis treatments. Our objective was to assess the topical treatment of chalcones and their combinations. The C33-A cell line has been extensively used for cytotoxicity evaluations. For instance, Sashidhara et al. (2010) utilized coumarin-chalcone hybrids and reported IC50 values ranging from 3.59 to 8.12 μM, with the most promising compound, 26, showing approximately 30 times higher selectivity for C33-A cells (doi.org/10.1016/j.bmcl.2010.10.116). Additionally, the HeLa cell line (human epithelial adenocarcinoma cells) is frequently used for cytotoxicity testing. Rathnakar et al. (2017) evaluated the cytotoxicity of chalcones and their flavonoid derivatives using HeLa cells, achieving their experimental objectives (DOI: 10.4172/2161-0444.1000480). Other relevant studies include those by Rathod et al. (https://doi.org/10.1002/aoc.7270) and Majumder et al. (DOI: 10.1007/s00203-022-03147-7). Therefore, the use of the C33-A cancer cell line was deemed the most appropriate choice for this research.
Also, the requested corrections have been made. The concentration of 4.69 mg/L was initially described as it was not statistically different from the negative control. However, it has been clarified that 9.38 mg/L was the highest non-cytotoxic concentration obtained for both substances, with cell viability above 70%, in accordance with the ISO EN 10993–5 protocol. Consequently, the text has been revised for greater clarity.Finally It was described in more detail in the caption for Figure 2 in Line 347 – 355.
2) Please state what statistical test was used in each experiment and add this information in the figure and table captions.
Answer: One-way ANOVA followed by Bonferroni analyzes were used and this information was added in the captions.
3) The discussion should be strengthened because in the chapter combining the results and discussion, the obtained results and their interpretation are mainly described. However, there is no discussion in the context of other studies
Answer: A discussion-only topic was created discussing the results point by point.
4) Please check the unit of concentration of the tested substances throughout the work, because sometimes it is given in mg/mL, sometimes in mg/L - it should be standardized.
Answer: mg/L was standardized throughout the text.
5) Definitions of the values used should be provided, e.g. SI, FICI, MFC and so on.
Answer: All the values end acronyms have been defined
6) In materials and methods, point 2.1, the authors write that they used three drugs: FCZ, CTZ, AMB, while in point 2.2.1, lines 129-130, only FCZ and AMB were used to assess the susceptibility of Candida spp.
line 170 has been corrected
Why was a different FCZ concentration range used for C. krusei than for other Candida? However, in Table 1, the MIC and MFC results of all preparations and drugs are given for all strains. Please check and correct the information in point 2.2.1.
Answer: The concentration range tested for C. krusei differs due to this species' higher resistance to FCZ compared to the other species used. According to the M27-A2 protocol (CLSI), C. krusei is the species for which the FCZ concentration range encompasses the highest levels. Different ranges are necessary due to the varying susceptibilities of the species.
7) Please explain why the antifungal activity of the combined preparations and drugs was assessed only against three out of five Candida species - C. albicans, C. tropicalis, C. krusei. On what basis were they selected? Moreover, why was AMB not used in these studies?
Answer: The selection of species for evaluating combinatory antifungal activity was based on the prevalence of C. albicans and the intrinsic resistance of C. krusei and C. tropicalis to azole drugs among non-albicans species. Given the focus on potential applications against vulvovaginal candidiasis, fluconazole was chosen as the drug, as it is one of the drugs of choice for treating VVC and is also a control drug according to the CLSI guidelines. Therefore, our research group decided to use fluconazole in these assays.
8) Table 4 is the EC50 of 2-HC against the three strains the same as in the case of 3'-OH? Whether different, please indicate this clearly.
Answer: The acronym EC was replaced by IC (inhibitory concentration 50%). These data are in table 2 now.
9) line 284 what does CIM80 mean? Shouldn't it be MIC80?
Answer: Correct, it has been corrected
10) explanations of abbreviations should be placed below the table and not in the title
Answer: The requested correction was made.
11) line 267 - standardize the letter size - uppercase or lowercase, check it out throughout
Answer: The requested correction was made.
12) line 221 please complete the method description
Answer: The method description has been rewritten as requested Line 307
13) lines 256-263 the authors describe the results presented in Table 1, therefore line 257 does not present the results from Table 1 but the concentration range used to determine MIC and MFC - please correct. Line 257 this description is incorrect because the table only gives established MIC and MFC values ​​for individual Candida species, and not the range of concentrations used, please change it
Answer: The requested correction was made in Line 230-237.
14) why the infected insects were given such a high dose of 350 mg/L FCZ (line 231) and Fig 6. On what basis was this concentration chosen, because previously Fig 5 was administered 0.13 mg/L
Answer: The figure 5 is now figure 6 (line 517) and show the results of the toxicological assays, and Figure 6 is now figure 7 (line 548) and presents the therapeutic efficacy outcomes of the compounds following infection with C. albicans. The application of FCZ at 350 mg/mL serves only as a control group for treating infected larvae and was not used in the toxicity assays. The treatment control group consists of larvae infected and treated with established effective doses for comparison with the proposed treatments. This concentration was chosen based on the scientific article: https://doi.org/10.1128/aac.00895-17. In this study, the authors describe dosage strategies of 15 to 20 mg/Kg. Our research group adopted a value of 17.5 mg/Kg for our experiments, equivalent to 350 mg/L according to the larval weights used. This reference has been added to the manuscript.
A dose of 0.13 mg/mL of FCZ was administered as this was the minimum inhibitory concentration found in our study for FCZ against C. albicans. The larvae received the combinations at the MIC values for the combinations and the MIC values for the individual substances.
please explain the difference between MIC (Table 1) and MIC80 (Table 2) as their values ​​are different in the tables. If MIC values were determined for all species, they should be shown for all strains tested. On what basis were only three species C. albicans, C. tropicalis and C. krusei selected for combinatorial testing of antifungal activity? This needs to be justified and why wasn't AMB used in this test if it worked at much lower concentrations than FCZ and CTZ? on what basis were the doses selected for in vivo cytotoxicity testing, shown in fig5E, F
15) line 259 - MIC 3'-HC for C. albicans is 3.25, and according to Table 1 is 31.25 mg/ml - please correct and insert the correct value
Answer: The requested correction was made line 333.
16) line 267 explanation of the abbreviations MIC and MFC standardize the letter size - uppercase or lowercase, this applies to the entire text please check and correct it
Answer: The requested correction was made in line 342.
17) I have reservations to the control in the experiment, because if the groups tested after C. albicans infection were given in a combination of 3’-HC+2-HC with different concentrations or FCZ, i.e. the insects were subjected to double immunization. Why were control larvae immunized only once if the study groups were immunized twice? To correctly interpret the results, you must prepare the appropriate controls.
Answer: All larvae, both in the control and test groups, received a single injection in the last proleg. The combinations were prepared in PBS containing the two active principles, and the final solution was injected into the larvae. All groups were subjected to the same experimental conditions, with identical volumes and quantities of injections. (line 306)
18) Why, for in vivo toxicity testing on Galleria mellonella larvae, much lower concentrations of 3'-HC and 2-HC were used alone than those administered in combination. On what basis were they selected for experiments? How much LC (1) and LC (2) were administered? How was the dose estimated? On what basis was the dose of C. albicans selected for insect infections? - please complete the information. Why is the health index not shown on day 5 but on day 4, since percent survival was assessed for 5 days. How many times were experiments on insects repeated? There is no statistical evaluation of the obtained results.
Answer: The isolated substances were administered at the minimum inhibitory concentrations obtained in the in vitro tests, as well as the associated substances, which were also administered at concentrations corresponding to the lowest FICI, however, as it is a more complex organism in relation to the in vitro tests, the doses were also administered. are transposed taking into account the larval weights and the pharmacokinetics of the substances in the larvae. The doses were determined based on the evaluation of the solubility of the substances. Because it is a very viscous formulation that we would have difficulty injecting, 0.25 g of LC was dissolved in 200 uL of PBS and 10 uL was administered. The intention was to dissolve the LCs in the smallest possible volume of PBS just to facilitate application. The concentration of C. albicans used was based on this scientific article: SCORZONI, L.; DE LUCAS, M. P.; MESA-ARANGO, A. C.; FUSCO-ALMEIDA, A. M.; LOZANO, E.; CUENCA-ESTRELLA, M.; MENDES-GIANNINI, M. J.; ZARAGOZA, O. Antifungal efficacy during Candida krusei infection in non-conventional models correlates with the yeast in vitro susceptibility profile. PloS one, v. 8, no. 3, p. e60047, 2013. DOI 10.1371/journal.pone.0060047.
The methodology carried out by our research group includes the 2nd and 4th days. The experiments were carried out in duplicate and with n=8 in each group. Statistical evaluation was added. Lines 306 – 325.

Reviewer 3 Report
Comments and Suggestions for Authors
This study aimed to explore two hydroxychalcones and their associations with azole drug activity against Candida species and the toxicological safety of, as well as to develop a lipid carrier capable of solubilizing and transporting these molecules with low aqueous solubility. Overall, this study is well-designed and their findings were significant. Thus, I just have two minor suggestions.
1. Please add some detail data in the result part of abstract.
2. Please separate result and discussion section.
Author Response
Revisor 3
- Please add some detail data in the result part of abstract.
Answer: It was added
- Please separate result and discussion section.
Answer: It was separate Line 555.
Round 2
Reviewer 1 Report
Comments and Suggestions for Authors
Thank you for addressing my concerns, I have no further suggestions.
Author Response
Prezado revisor, Muito obrigado pelas correções e sugestões.Reviewer 2 Report
Comments and Suggestions for Authors
I would like to thank the authors for responding to the comments and explaining them, as well as correcting the article.
Nevertheless, I still have some minor suggestions e.g.,
- the authors write that they assessed the insects for 5 days, but there is no information whether the insects were provided with food during this time. It is known that this has a very large impact on the immune response and survival
- I understand the reason for working on the C33 cell line, but I still believe that cytotoxicity tests should be carried out on the correct cell lines because their effects do not necessarily have to be neutral for them
-it is worth including information in the text (e.g. results or discussion) why the C33-A human cervical cancer cell line was chosen to determine cytotoxicity
- full Latin names are usually given only for the first time in the text,
and then the abbreviation Candida albicans - C. albicans, or Galleria mellonella -
G. mellonella please follow this throughout the text (e.g. lines 61, 152, 256, 400,
431, 442, 476, 547 etc.)
- line 127 is "...at 37°C and 24 hours.", would be better ''for 24 hours''
- please correct the markings in the photo in Figure 3 because it does not match the description
- please correct the title of the results subsection, line 398, instead of "(Galleria mellonella)" in brackets, better e.g." using Galleria mellonella"
Author Response
The manuscript's authors thank the reviewers for their careful review and suggestions.
Reviewer 2
- the authors write that they assessed the insects for 5 days, but there is no information whether the insects were provided with food during this time. It is known that this has a very large impact on the immune response and survival.
Answer: As requested, the correction has been made and highlighted in yellow in the manuscript.
- I understand the reason for working on the C33 cell line, but I still believe that cytotoxicity tests should be carried out on the correct cell lines because their effects do not necessarily have to be neutral for them
-it is worth including information in the text (e.g. results or discussion) why the C33-A human cervical cancer cell line was chosen to determine cytotoxicity
Answer: Firstly, we understand the importance of using primary cell lines. However, it is also important to remember that in some research centers, the availability of these cell types is not easily accessible due to the need for them and funding limitations. The C33A line was chosen because it is available in the laboratory and is easy and cheap to cultivate, like other study strains. Furthermore, the cytotoxicity test was planned in this study to preview the toxicity test in a more complex model, which is G. mellonella. Thinking about the principles of the 3R's currently discussed in the research, the methodology used in this study, utilizing the lineage before the animal (even though this strain is tumorous), contributes to an insight that consequently leads to less use of animals. Another point that still needs to be highlighted is that although C33A is tumoral, it represents the target anatomical site of the study (vaginal epithelium). However, the use of this cell line for evaluating chalcones and other substances in terms of cytotoxicity is also described in the literature (DOI: 10.1002/ardp.201800295; DOI: 10.2174/1573406415666190724145158).
- full Latin names are usually given only for the first time in the text, and then the abbreviation Candida albicans - C. albicans, or Galleria mellonella - G. mellonella please follow this throughout the text (e.g. lines 61, 152, 256, 400, 431, 442, 476, 547 etc.)
Answer: Corrections have been made, which are highlighted in yellow in the manuscript as requested.
- line 127 is "...at 37°C and 24 hours.", would be better ''for 24 hours''
Answer: As requested, the correction has been made and highlighted in yellow in the manuscript.
- please correct the markings in the photo in Figure 3 because it does not match the description
Responder: Conforme solicitado, a correção foi feita e destacada em amarelo no manuscrito.
- por favor, corrija o título da subseção de resultados, linha 398, em vez de "(Galleria mellonella)" entre parênteses, melhor, por exemplo, usando Galleria mellonella"
Responder: Conforme solicitado, a correção foi feita e destacada em amarelo no manuscrito.
